# Modeling Analysis of SM2 Construction Attacks in the Open Secure Sockets Layer Based on Petri Net

**DOI:** 10.3390/s22041398

**Published:** 2022-02-11

**Authors:** Xi Deng, Liumei Zhang, Yichuan Wang, Fanzhi Jiang

**Affiliations:** 1School of Computer Science, Xi’an Shiyou University, Xi’an 710065, China; 20222060785@stumail.xsyu.edu.cn (X.D.); 19211060559@stumail.xsyu.edu.cn (F.J.); 2School of Computer Science and Engineering, Xi’an University of Technology, Xi’an 710048, China; chuan@xaut.edu.cn; 3Shaanxi Key Laboratory for Network Computing and Security Technology, Xi’an 710048, China

**Keywords:** Petri net, Petri net model, CVE-2021-3711 vulnerability, cybersecurity

## Abstract

The detection and defense of malicious attacks are critical to the proper functioning of network security. Due to the diversity and rapid updates of the attack methods used by attackers, traditional defense mechanisms have been challenged. In this context, a more effective method to predict vulnerabilities in network systems is considered an urgent need to protect network security. In this paper, we propose a formal modeling and analysis approach based on Petri net vulnerability exploitation. We used the Common Vulnerabilities and Exposures (CVE)-2021-3711 vulnerability source code to build a model. A patch model was built to address the problems of this model. Finally, the time injected by the actual attacker and the time simulated by the software were calculated separately. The results showed that the simulation time was shorter than the actual attack time, and ultra-real-time simulation could be achieved. By modeling the network system with this method, the model can be found to arrive at an illegitimate state according to the structure of Petri nets themselves and thus discover unknown vulnerabilities. This method provides a reference method for exploring unknown vulnerabilities.

## 1. Introduction

With the development of society, the Internet plays an increasingly significant role in our daily lives. However, there are numerous attackers on the network who try to use it to perform illegal operations, such as illegal access, data leakage, data overwriting, etc. [1]. These actions can generate chaos in society and cause economic damage or even the loss of life. Therefore, protecting network security has become a very vital issue.

In recent years, a number of solutions have been proposed around network security. Protecting network security can focus on the following aspects: physical measures: safeguarding essential network equipment (such as switches, large computers, etc.), developing rigorous rules and regulations for network security, and taking measures such as radiation protection, fire prevention, and installation of uninterruptible power supplies; access control: strict authentication and control of user access to network resources; data encryption: this method keeps data secure and provides guaranteed secure deletion by data encryption, which uses the identical key to encrypt all plain text and stores this key in a separate block. It is also possible to combine machine learning algorithms with security to protect information security [2]. The article takes a look at the vulnerabilities of the system to find the location of the problem and repair it.

A vulnerability flaw in hardware, software, protocol implementation, or system security policy allows an attacker to gain unauthorized access and compromise the system. Buffer overflow injection is one of the common and dangerous vulnerabilities among a number of security bugs. A buffer overflow vulnerability was recently disclosed in the National Vulnerability Database called CVE-2021-3711. When a computer pads a buffer with more bits than it can hold, this overwrites legitimate data. Ideally, the system would check the data size and not allow characters that exceed the length of the buffer to appear. However, most systems assume that the data width always matches the allocated storage space, which creates the conditions for implementing the buffer overflow.

Common Vulnerabilities and Exposures (CVE) is a publicly available list of computer security vulnerabilities. The CVE program is administered by MITRE Corporation and funded by the Department of Homeland Security’s Cybersecurity and Infrastructure Security Agency. The CVE content is brief and does not contain technical data or information about risks, impacts, and remediation. These details appear in other databases, including the U.S. National Vulnerability Database (NVD), the Computer Emergency Response Team/Coordination Center Vulnerability Description Database, and various lists maintained by vendors and other organizations. A vulnerability is assigned a CVE ID when it meets three specific criteria: it can be independently remediated; it is identified or documented by the affected vendor; it affects a codebase. In addition, CVE vulnerabilities have scoring systems to assess the severity of the vulnerability. One is the Common Vulnerability Scoring System (CVSS), a set of publicly available criteria for assigning numbers to vulnerabilities to assess their severity. CVSS scores are used by the NVD, CERT, and other agencies to assess the impact of vulnerabilities. Scores range from 0.0 to 10.0, with higher numbers indicating a higher severity of the vulnerability. Vulnerabilities scoring 7.0 to 10.0 are typically considered high risk, those scoring 4.0 to 6.9 medium risk, and those scoring 0.0 to 3.9 low risk.

In August 2021, the NVD released the details of a vulnerability called CVE-2021-3711. The flaw had a CVSS 3.1 score of 8.1, which is a critical vulnerability. A malicious attacker provides the SM2 content for decryption to the application causing the attacked data to overflow the buffer, thereby changing the contents of other data stored behind the buffer. The vulnerability affects all versions of the Open Secure Sockets Layer (OpenSSL) prior to 1.1.1l, which contain the SM2 quotient algorithm.

In the case of vulnerabilities that have already appeared, the attack process can be reproduced. Consider the details of the attack from the viewpoint of the attacker, and find the vulnerability points of the system to be patched. However, predicting unknown vulnerabilities and minimizing attackers’ attackable points is a direction in the field of cyber security. In the paper, we take the modeling from the CVE-2021-3711 vulnerability and find the problems. This modeling approach applied to systems or data streams provides the possibility of predicting unknown vulnerabilities.

The Petri net modeling approach was chosen because of its inherent advantages to abstract the vulnerability exploitation process. Intuitive modeling: Petri nets have the ability to describe asynchronous concurrency with its graphical representation. the structure of the Petri net mesh produces a partial order that makes describing asynchronous concurrency possible, and the graphical representation is more consistent with the reality of asynchronous concurrency. In addition, the token representing the Petri net token configures the distributed state of the system, which informs the current dynamics of the whole system. Theoretical support: The Petri net modeling approach is simulated according to the vulnerability exploitation process. Because it does not involve the processing of actual data, the model processing speed may be faster in most cases, thus providing theoretical decision support for system vulnerability detection and effectively improving network security.

In this study, we aimed to analyze network vulnerabilities through Petri nets with the following contributions:(1)Modeling the vulnerability exploitation process using Petri nets: In contrast to other research works, such as the SQL injection attack analysis method proposed in the study [3], depicting the attack process does not involve the underlying principles. Therefore, in this paper, Petri nets were used to model the attack process from the source code, and the model depicts the attack process in more detail;(2)System vulnerability detection: The accessibility of nodes in the Petri net model was used to provide theoretical support for detecting unknown vulnerabilities and generating vulnerability code;(3)Experimental evaluation: In this paper, the time of the actual attack and the time when the model first reaches the insecure state were counted in 10 groups, respectively. We calculated their average time. The experimental results showed that the time of the model simulating the attack was shorter than that of the actual attack and can achieve ultra-real-time simulation.


This paper is organized as follows. Section 2 introduces work related to this paper. A Petri net model of a vulnerability with CVE-2021-3711 is proposed in Section 3. Section 4 presents the model analysis, including SM2 ciphertext and ASN.1 encoding, the Petri net model analysis of CVE-2021-3711, and the Petri net model with patch analysis of CVE-2021-3711. Section 5 provides the experiment analysis, including the model analysis and an example study. Conclusions and future work are proposed in Section 6.

## 2. Related Work

With the rapid development of computer network technology, people are increasingly aware of the importance of network security. In this context, the research community has proposed the analysis of network security from different perspectives. For instance, the work in [4] proposed a new method to represent computer networks and intrusion detection systems. Such an approach can effectively simulate network attack scenarios to test and evaluate network security systems. The work in [5] presented an attack-trees-based approach to information system security analysis. The attack trees contain social engineering attacks and attacks that require physical access control areas. The network security analysis method based on software technology attacks was extended. The researchers in [6] proposed a security situational awareness system with high real-time capacity and accuracy in security trend prediction. The system satisfies the requirements for the analysis and prediction of large-scale network security conditions. The study in [7] proposed a traffic analysis tool. The tool was designed to provide scalable analysis and services for network traffic data, allowing attackers to explicitly engineer their actions or hide attacks within the broader normal activity, thereby improving network visibility and security. The work in [8] developed a model to predict the timing of potential attacks. Such a model predicts future malicious attacks with an average accuracy of 94.9% within a week by analyzing the structural risk level of the malware distribution network, the connectivity of the malware in question, and the timing of the malware.

However, vulnerabilities are discovered on computers so frequently that system administrators are not immediately capable of patching all these flaws on hosts within the network. Vulnerability analysis and detection are an effective way to counteract malicious attacks and prevent them from committing harmful behaviors. In this context, researchers have proposed detection techniques and analysis methods for different vulnerabilities. The researchers in [9] proposed a machine learning approach taxonomy to detect software vulnerabilities. Machine learning includes supervised learning, semi-supervised learning, and deep learning. The work in [10] proposed a slice-based intelligent detection system for binary code vulnerabilities. It can effectively improve the accuracy of binary vulnerability detection. In the study of [11], a framework was extracted to detect and prioritize attacks without patches. This framework uses probabilities to identify attack paths and classify vulnerabilities. In [12], a neural-network-based approach was proposed to distinguish vulnerabilities. Due to the inconsistency of information between attackers and targets, vulnerabilities can be classified into known and unknown vulnerabilities. Existing research has focused on the risk assessment of known vulnerabilities. However, unknown vulnerabilities are more threatening and harder to detect.

To overcome the limitations of known vulnerability analysis, many analysis techniques for unknown vulnerabilities have been developed. The study in [13] showed a solution for unknown vulnerability risk assessment based on directed graphs, which fills the current gap in this research direction of investigation. In [14], the authors presented a network hardening approach that gives a maximized security solution for unknown and unfixable vulnerabilities by unifying the hardening options under the same model. The researchers in [15] extracted a vulnerability syntax tree to predict unknown vulnerabilities in software applications. The experimental results showed that the prediction rate, recall rate, and accuracy rate were significantly improved.

We recall the basics of Petri net. The concept of Petri nets was first proposed by Carl Adam Petri in his doctoral thesis in 1962. Since the first international seminar on Petri net theory and application was held, the theory and application of Petri nets have been continuously enriched and improved. The main analysis methods of the Petri net model depend on: reachable tree, correlation matrix and equation of state, invariants, and analysis simplification rules. Its vertical development is shown as follows: from the basic event network, through the location change network, to the advanced network. Its horizontal development is as follows: from the net without parameters to the time Petri net and stochastic Petri net; from a general directed arc to a forbidden arc and a variable arc; from the number of natural number markers to the number of probability markers; from atomic transition to predicate transition and subnet transition.

Triple N=(S,T;F) presents a network structure. *S* is a finite set of places. *T* is a finite set of transitions. *F* is a set of arcs, which are from the places to the transitions and from the transitions to the places. The union of *S* and *T* is not empty, and the intersection of *S* and *T* is not empty. $\Sigma =(S,T:F,K,W,M_{0})$ is a Petri net system. *S*, T, and *F* have the same meaning as above. *K* is the capacity function on the directed network. *W* is the weight function on the directed network *N*. $M_{0}$ is the initial recognition [16].

Many research methods have been based on Petri nets. For instance, the study in [17] proposed a reward and penalty trust model for performance evaluation. The models explain situations better than the classical models that are commonly used. The researchers in [18] presented a framework based on Petri nets to analyze structural security in e-business processes. This framework helps professionals study whether the structure of e-business processes is secure. In [19], the author extracted a timed colored Petri net to simulate security failure probability analysis. In view of the influence of different numbers of attackers on the attack time, the corresponding modeling method based on time color Petri net was presented in this paper. The security failure probability can be obtained by comparing the duration of the attack process with the assumed security check interval. The work in [20] proposed an integrated Bayesian-Petri net method to quantify individual influence in domino effect scenarios. Research can protect resources and minimize individual lives at oil and gas facilities.

Meanwhile, the modeling approach of Petri nets can be applied to network security vulnerabilities. Wu R et al. [21] proposed a modeling approach based on hierarchical colored Petri nets to simulate multi-stage attacks, which helps to understand the specific attack process and take effective protection measures. Wang L et al. [22] proposed an approach based on colored Petri nets, which investigates cache attacks and security mechanisms. The results showed that this modeling approach can evaluate and compare the threat level of cache attacks in computer environments with different security mechanisms. In this study, the attack procedure of buffer overflow vulnerability CVE-2021-3711 was analyzed by Petri nets for the purpose of maintaining network security.

In order to build a more detailed model, each major step in the source code of CVE-2021-3711 corresponds to a place or a transition of the Petri net model. The model sets the initial token value and the maximum capacity of each place. The attack procedure can be simulated by the Tina software. The time of the actual attack is calculated and compared with the time of the actual attack.

## 3. Petri Net Model of a Vulnerability with CVE-2021-3711

In this section, the article describes a model based on CVE-2021-3711. The vulnerability affects all versions of OpenSSL before 1.1.1l that contain the SM2 algorithm. The reason for the existence of the flaw is that a chunk of memory is allocated when SM2 is decrypted, and the result of the decryption may be larger than the capacity of that allocated memory, resulting in an out-of-bounds memory write.

The model is based on the important steps of the CVE-2021-3711 attack process in Figure 1. Fourteen places in the figure are described as shown in Table 1 and eleven transitions as shown in Table 2. P1–P3 give the process of packet construction. P4 and P5 are the process of packet sending and receiving. P6–P9 are the process of packet decoding. The initial token value is set at P0. The token will go through T0 to P1, P2, and P3, respectively. The token of P2 can only go to one of T1 or T2. P1 and P3 contend for the token of P2, and only one occurs under the same time point of P1 and P3. The token arrives at T5 after P4 occurs in the sequence. P7 and P9 contend for the token of P8,one of which occurs at T6 or T7. The token can be recycled after reaching P12 and returned to P0; the token stops flowing after reaching P13. The flow of the token matches the attack process of CVE-2021-3711. In order to provide the readers with the possibility to reproduce the experiments of this paper, the download link is: https://github.com/funssie/ (accessed on 19 December 2021). This project contains the main Petri net model txt documents that appear in the article.

## 4. Models Analysis

### 4.1. SM2 Ciphertext and ASN.1 Encoding

CVE-2021-3711 decryption involves SM2 ciphertext and ASN.1 encoding. To better describe the relationship between the code and the model, this section describes the SM2 algorithm and the SM2 cipher text encoding of ASN.1.

SM2 is an elliptic curve public key cryptography asymmetric algorithm [23]. The algorithm can be divided into a public key and a private key. If the public key is given to others, it can be disclosed to a certain extent. If the private key is kept for yourself, it must be kept secret. The public key can be calculated from the private key; computing the private key from the public key is quite difficult, and at this stage, it is impossible to achieve confidentiality.

Suppose the message to be sent is a bit string M, and “klen” is the bit length of M. In order to encrypt plaintext M, User A, as the encryptor, should perform the following operations:-A1: Use a random number generator to generate random number k∈[1,n−1];-A2: Calculate the elliptic curve point C1=[k]G=(x1,y1) ([k]G stands for k∗G), and convert the data type of C1 to a bit string;-A3: Calculate the elliptic curve point S=[h]PB; if S is an infinite point (*h* is a cofactor of *n*), then output error and exit;-A4: Calculate the elliptic curve point [k]PB=(x2,y2), and convert the data type of coordinates x2 and y2 to a bit string;-A5: Calculate t=KDF(x2‖y2,klen); if t is an all 0 bit string, return A1 (KDF is the key derived function);-A6: Calculate C2=M⊕t;-A7: Compute C3=Hash(x2 ‖ M ‖ y2);-A8: Output ciphertext C=C1 ‖ C2 ‖ C3.

Then, we suppose klen is the bit length of C2 in the ciphertext. To decrypt the ciphertext C=C1 ‖ C2 ‖ C3, User B as the decryptor should perform the following operations:-B1: Take out the bit string C1 from *C*, and convert the data type of C1 to the point on the elliptic curve, then verify whether C1 meets the elliptic curve equation; if not, an error will be reported and exit;-B2: Calculate the elliptic curve point S=[h]C1; if *S* is an infinite point, then error and exit;-B3: Calculate [DB]C1=(x2,y2), and convert the data type of coordinates x2 and y2 to a bit string;-B4: Calculate t=KDF(x2 ‖ y2,klen); if t is an all 0 bit string, then error and exit;-B5: Take the bit string C2 from *C*, and calculate $M^{\prime }=C2⊕t$;-B6: Compute $u=Hash(x2\;\|\;M^{\prime }\;\|\;y2)$, and extract bit string C3 from *C*; if u/C3, an error is reported and exit.-B7: Output plaintext $M^{\prime }$.

The general form of ASN.1 encoding is reported in Table 3. The type flag bit occupies one byte, and the load length can be short code or long code. Therefore, the load length should be at least one byte and may be longer. The payload may be the same length as the data to be encoded, or it may include padding bytes.

Therefore, according to GB/T 35276-2017, the SM2 ciphertext needs to be ASN.1 encoded as follows:



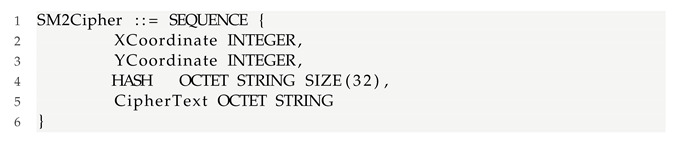



HASH is an SM3 HASH value with a fixed length of 32 bytes, and CipherText is the ciphertext corresponding to the plain text. Figure 2 shows five pairs of type flag bit, length of load. For ASN.1 encoding of SM2 ciphertext, the minimum sum of their lengths is 10 bytes. Each load length occupies at least one byte, and may exceed one byte in practice.

### 4.2. Petri Net Model Analysis of CVE-2021-3711

In this subsection, we analyze the Petri net model of CVE-2021-3711 shown in Figure 1. Fourteen places, eleven transitions, and the connection between places and transitions are called to the Petri net structure. The reason for the vulnerability in CVE-2021-3711 is that when decrypting SM2-encrypted data, the memory allocated is less than the memory needed for the actual decryption of the plaintext, resulting in buffer overflow. Therefore, the model in this article focuses on the state of the system, the construction of SM2 packets, and buffer allocation.

First, the normal state of OpenSSL is represented by P0, and if an attacker attempts to inject, it is represented by T0. Then, the attacker needs to carefully construct SM2 packets to achieve buffer overflow. The process of packet construction is represented by P1, P2, and P3. Illegal and legal completed construction packets are represented by T1 and T2, respectively.

Next, the attacker will send the constructed packet, and the status of the packet is represented by P4. The receiver determines whether the SM2 packet is represented by T3. Then, the state of receiving packet is represented by P5.

Now, decrypt the SM2-encrypted data, and the system needs to call the API function EVP_PKEY_decrypt(). Typically, an application will call the function twice during decryption. The first time, the “out” parameter can be NULL, and the “outlen” parameter is populated with the buffer size required to hold the decrypted plaintext. The application can then allocate a sufficiently sized buffer.



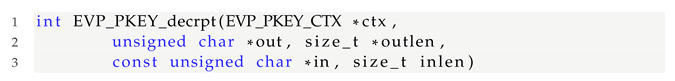



The second time, the “out” parameter can be the starting address of the newly assigned buffer. When the “out” parameter cannot be NULL, the system calls the function SM2_plaintext_size() to perform the decryption operation.



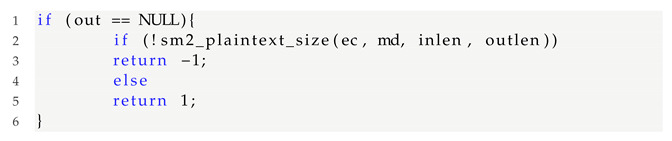



Next, the lengths of the x coordinate and y coordinate are represented by 2 ∗ field_size. The length of the SM3 hash value is represented by md_size. The minimum encoding overhead of ASN.1 is 10 bytes, as mentioned above. The ∗pt_size parameter is always used to indicate the length of the plaintext obtained after decryption. The value of the md_size variable is constantly 32 bytes. Since the coordinates of a point on an elliptic curve of the SM2 algorithm are usually represented by two 32-byte coordinates (x,y), the length occupied is generally 32 bytes.



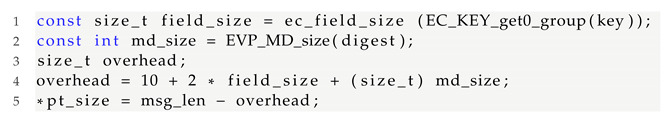



Therefore, we suppose vectors *X* and *Y* are 32 bytes, respectively. This operation is represented by T4. At this time, the system is in the ready state and is represented by P6. Next, the system allocates memory represented by T5. The state of the successfully obtained buffer is represented by P8. The state of the obtained buffer size is represented by P7, which is consistent with the actual plaintext size after decryption, whereas P9 indicates inconsistency. Next, we use T6 and T7 to represent legal and illegal actions, respectively. P10 and P11 indicate the buffer security status and buffer overflow status, respectively. In this case, attacker injection failure is represented by T8, system security by P12, and token recovery by T10. Similarly, the successful injection of the attacker is denoted by T9, and the insecure state of the system is denoted by P13.

### 4.3. Patch Analysis of CVE-2021-3711

In this subsection, we illustrate the Petri net model by considering the patch analysis of CVE-2021-3711 shown in Figure 3. A bug in the implementation of the SM2 decryption code means that the calculation of the buffer size required to hold the plaintext returned by the first call to EVP_PKEY_decrypt() can be smaller than the actual size required by the second call. Table 4 shows the meaning of places. Table 5 shows the meaning of the transitions.

Therefore, in the fixed code for Version 1.1.11, the SM2 plaintext length calculation has been modified. First, the SM2 ciphertext in ASN.1 encoding form is decoded.



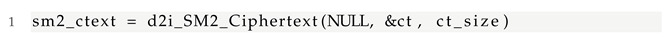



Assign the SM2 plaintext length obtained at decoding to the parameter ∗pt_size. At last, the system releases the buffer dynamically opened inside the decoding function d2i_SM2_Ciphertext() when it is called.



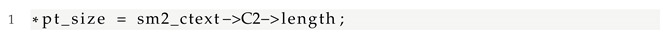



After the restoration, the calculation of the buffer size required to hold the plaintext returned can be equal to the actual size required. Therefore, the decoded cipher is represented by T4, and the values of the resolved vectors *X* and *Y* are represented by P6 and P7, respectively. We set the values of *X* and *Y* to be the same as the actual values, which is represented by T5. The parts marked with blue dotted lines are patching operations. Since, in a patched model, the bytes of the vectors *X* and *Y* must be decoded, the probability P=1 is used to represent what is certain to happen. This operation without decoding will not happen, which is represented by T8. For the sake of the experiment, P=0 is used to represent what is not happening. The red arrows are unreachable. The system is always safe.

## 5. Experiment Analysis

### 5.1. Model Analysis

In this section, this paper counts the time when the Petri net model first arrives at a secure state, the time when it first arrives at an insecure state, and the time of the actual attack, respectively. The corresponding version numbers of the system environment, the software for the model simulation, and the environment for the verification experiments that will be used to complete the experiments are shown in Table 6.

First, Tina builds models and performs simulation experiments. The Petri net model is used to reach a safe state for the first time, as shown in Figure 4. In this model, P0 is considered to be the beginning of the model. The value of P0 represents the number of injections attempted by the attacker. We set the value of P0’s token to 10.

When the model starts to simulate and the token value of a place is greater than one, the number is displayed directly. The value of the token is equal to the number of black dots displayed by the places. Red transitions mean it has the right to occur. There are many transitions in the entire model, and the software is designed to perform randomly, so the total time to first reach the safe state P12 is random.

Similarly, we set the value of P0’s token to 10. Figure 5 shows the Petri net model used to reach an unsafe state for the first time.

The firing duration can be set to 0 ms, 1 ms, 10 ms, 100 ms, or 1000 ms. The firing duration means the speed of token flow in the model. The smaller the time, the faster the token will run. We set the firing duration to 1 ms and selected the untimed option. Figure 6 shows ten groups of experiments. The average time of the first arrival of the safe state and unsafe state was 31.84 ms and 30.042 ms, respectively.

This paper also counted how long it took an attacker to complete an attack. This time included the time T1 when the packet was transferred back and forth, the time T2 when the system allocates the buffer, the time T3 when the packet is processed, and the time T4 when the packet is put into the buffer. We counted the time to construct 200-byte and 800-byte packet injection, respectively. Figure 7 shows 10 groups of data, and the average time was 40.8041 ms and 43.6968 ms, respectively. The results showed that the running time of the model simulation was less than that of a real attack, and the simulation environment can realize faster-than-real-time simulation.

In Table 7, a fine-grained qualitative analysis and comparison of several typical vulnerability models are presented.The model proposed by Flammini F et al. [24] is a simple generic model based on stochastic Petri nets from which we can find the location of vulnerabilities. The model incorporates code for analysis and can be simulated using software. The model developed by Padilha B et al. [25] analyzes the vulnerability at the code level. Shen J et al. [26] built a model that can be simulated in software, but did not go up to the code analysis. The models proposed in Fodor K et al. [27] and Liu X et al. [28] can predict unknown vulnerabilities. However, the paper did not address the code for analysis. In general, the existing studies have not provided a fine-grained description of the vulnerability exploitation process and the analysis of patch models. This paper implemented a vulnerability exploitation mapping to a formal model from a micro perspective of the CVE-2021-3711 attack scenario. The details of the vulnerability analysis are extended to compensate for the shortcomings of existing approaches.

### 5.2. Illustrative Example

In this subsection, the paper illustrates the proposed vulnerability by considering ASN.1 decode. These data came from: https://fossies.org/linux/openssl/test/recipes/30-test_evp_data/evppkey.txt (accessed on 19 December 2021). The set of data was used to test that the y coordinate was less than 32 bytes.



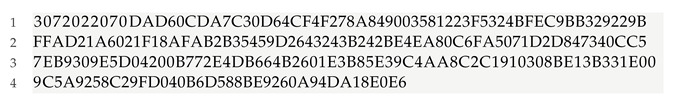



The length of the ciphertext is 116 bytes. Decrypt this set of ciphertexts in ASN.1 format as mentioned above and shown in Table 8.

The values of x and y can be obtained from the above analysis. b and p are constants. Equation (Equation 1) shows that the X coordinate and Y coordinate satisfy the SM2 elliptic curve. We verified that the elliptic curve equation was satisfied with Python in cmd:(1)$$y^{2}\%p==\left(x^{3}-3*x+b\right)\%p$$



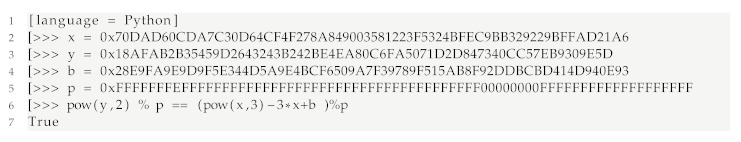



The first time the function pkey_sm2_decrypt() is called, the parameter “out” is NULL and parameter “msg_len” is 116 bytes. The function SM2_plaintext_size() returns 10 bytes. Then, the function OPENSSL_malloc() allocates 10 bytes of memory.The second time, since the ciphertext has 11 bytes, the function pkey_sm2_decrypt() results in 11 bytes. “Out” points to ten bytes of memory, while decryption results in eleven bytes, resulting in one byte overflow.

## 6. Conclusions and Future Work

In summary, this paper proposes a Petri net model for actual vulnerability exploitation. The buffer overflow vulnerability ID used in the experiment was CVE-2021-3711. We first analyzed the vulnerability’s underlying source code and proposed a Petri net model with better finesse. Then, we used the Petri tool Tina to conduct a dynamic simulation. The simulation result showed us that the experiment might reach an unexpected insecure state, proving our method is feasible for vulnerability detection and repair of the network system. Finally, we compared the time of actual attack and software simulation. It was clear that the time used to simulate the attack was much shorter, with a 26.37% reduction in time for 200 bytes, and the model was capable of faster-than-real-time simulation. Furthermore, by applying our formal modeling approach, we could observe whether the token would reach the insecure state in the simulation. Therefore, this lays a theoretical foundation in detecting unknown vulnerabilities in the network system.

There are similar patch sketches of the model, and only one of them was drawn in this paper. In addition, directions for further research include: (1) transforming the model into a matrix and extracting features from it; (2) multiplying many features of the vulnerabilities for aggregating into a graph. Furthermore, we may consider the use of the Petri net model so that unknown vulnerabilities can be defended against.

## Figures and Tables

**Figure 1 sensors-22-01398-f001:**
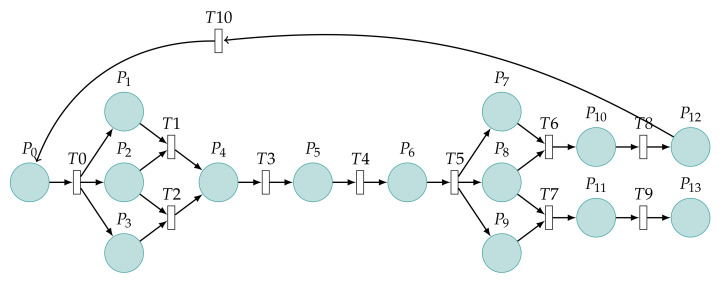
Petri net model of a vulnerability with CVE-2021-3711.

**Figure 2 sensors-22-01398-f002:**
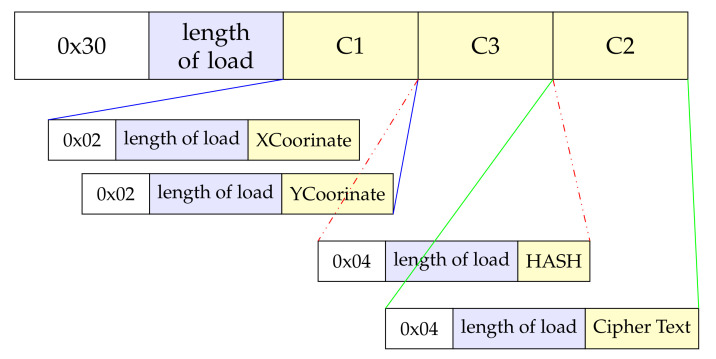
SM2 ciphertext encodes ASN.1.

**Figure 3 sensors-22-01398-f003:**
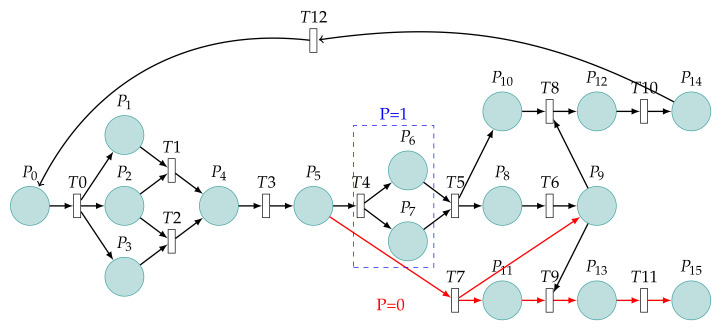
Petri net model with the patch.

**Figure 4 sensors-22-01398-f004:**
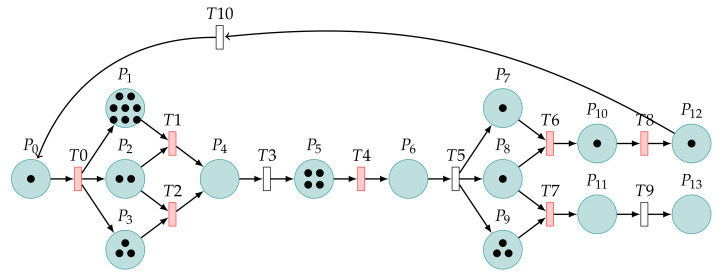
The model reaches a safe state for the first time.

**Figure 5 sensors-22-01398-f005:**
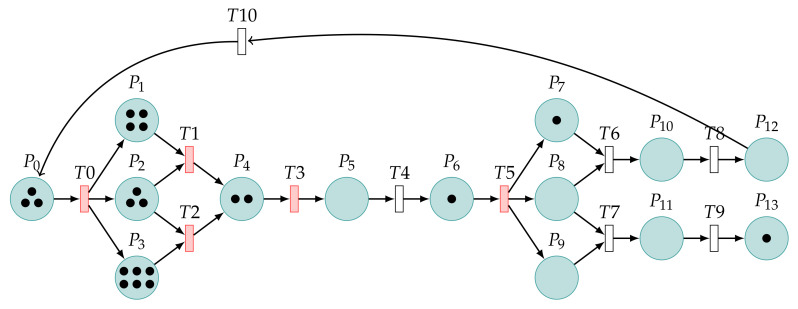
The model reaches an unsafe state for the first time.

**Figure 6 sensors-22-01398-f006:**
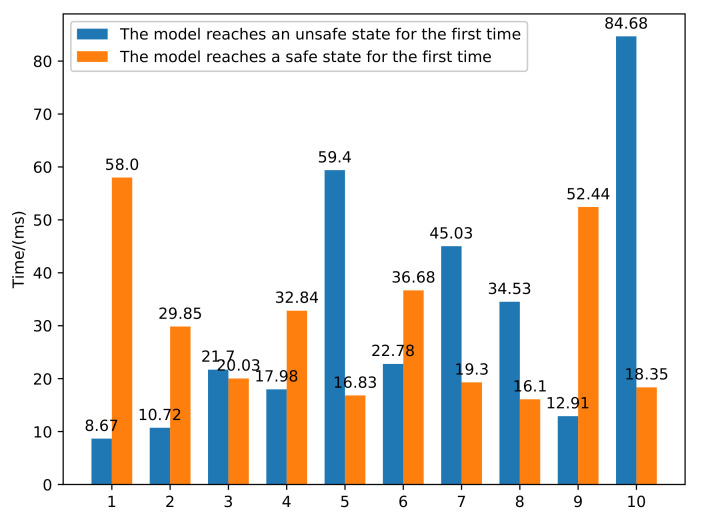
Simulation time statistics of the model.

**Figure 7 sensors-22-01398-f007:**
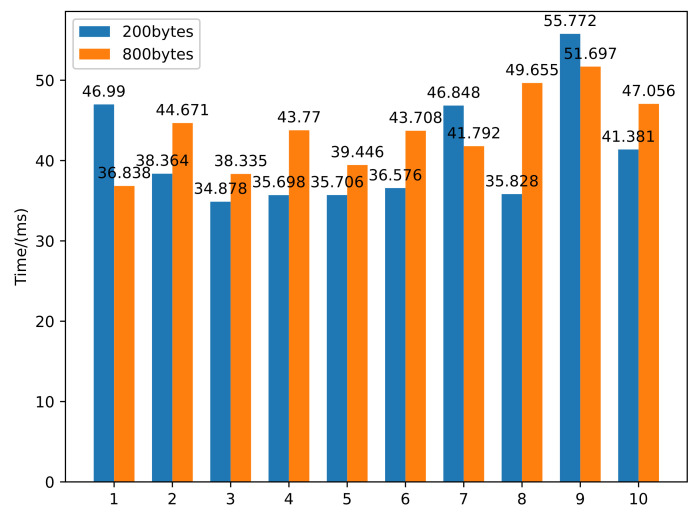
Simulation time statistics of the attack.

**Table 1 sensors-22-01398-t001:** The meaning of places.

Places	Description
P0	OpenSSL is a normal state
P1	Construct the SM2 packet legally
P2	Construct the SM2 packet state
P3	Construct the SM2 packet illegally
P4	Packets sent by the client state
P5	packets received by the receiver state
P6	The ready state
P7	Consistent state
P8	Get buffer state
P9	Inconsistent state
P10	Buffer security state
P11	Buffer overflow state
P12	System security state
P13	System insecurity state

**Table 2 sensors-22-01398-t002:** The meaning of places.

Transitions	Description
T0	Attempted to inject
T1	Legal structure
T2	Illegal structure
T3	Check whether SM2 packets are sent
T4	Set X, Y = = 32 bytes
T5	Allocate memory
T6	Legal allocation
T7	Illegal allocation
T8	Injection of failure
T9	Injection of success
T10	Recycling token

**Table 3 sensors-22-01398-t003:** General form of ASN.1.

Type Flag Bit	The Length of the Load	Load
1 byte	Variable length	Variable length

**Table 4 sensors-22-01398-t004:** The meaning of places.

Places	Description
P0	OpenSSL is a normal state
P1	Construct the SM2 packet legally
P2	Construct the SM2 packet state
P3	Construct the SM2 packet illegally
P4	Packets sent by the client state
P5	Packets received by the receiver state
P6	Get the size of the X coordinate state
P7	Get the size of the Y coordinate state
P8	The ready state
P9	Get buffer state
P10	Consistent state
P11	Inconsistent state
P12	Buffer security state
P13	Buffer overflow state
P14	System security state
P15	System insecurity state

**Table 5 sensors-22-01398-t005:** The meaning of transitions.

Transitions	Description
T0	Attempted to inject
T1	Legal structure
T2	Illegal structure
T3	Check whether SM2 packets are sent
T4	Decode
T5	Set X and Y equal to the actual value
T6	Allocate memory
T7	Do not decode
T8	Legal allocation
T9	Injection of failure
T10	Injection of success
T11	Recycling token

**Table 6 sensors-22-01398-t006:** The environment of the experiment.

Environment	Configuration
OS	Win10
CPU	Intel${}^{®}$ Core${}^{\mathrm{TM}}$ i7 – 10700
Tina tools	3.6.0
Python	3.8.6

**Table 7 sensors-22-01398-t007:** Result of the comparison with other studies.

Article	Finding the Location of the Vulnerability	Patch Model	Source Code Level	Software Simulation	Predict
Flammini F. et al. [24]	✓	×	✓	✓	×
Padilha B. et al. [25]	×	×	✓	✓	×
Shen J. et al. [26]	×	×	×	✓	×
Fodor K. et al. [27]	×	×	×	×	✓
Liu X. et al. [28]	×	×	×	×	✓
This article	✓	✓	✓	✓	✓

**Table 8 sensors-22-01398-t008:** Comparative analysis of the model.

Value	Meaning
30	Sequence type
72	the total length of subsequent data is 114 bytes
02	Integer type
20	X coordinate is 32 bytes
70DAD60CDA7C30D64CF4F278A8490035	32 bytes of X coordinate
81223F5324BFEC9BB329229BFFAD21A6	
02	Integer type
1F	X coordinate is 31 bytes
18AFAB2B35459D2643243B242BE4EA80	32 bytes of Y coordinate
C6FA5071D2D847340CC57EB9309E5D	
04	Octet string type
20	HASH is 32 bytes
0B772E4DB664B2601E3B85E39C4AA8C2	32 bytes of HASH
C1910308BE13B331E009C5A9258C29FD	
04	Octet string type
0B	Ciphertext is 11 bytes long
6D588BE9260A94DA18E0E6	11 bytes of ciphertext

## Data Availability

Not applicable.

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
