# Peer review of "Modeling Analysis of SM2 Construction Attacks in the Open Secure Sockets Layer Based on Petri Net"

_sensors, 2022, doi:10.3390/s22041398_

Round 1

Reviewer 1 Report

  1. The contributions are not clear. Could be explained in order to be interesting for readers

  1. A deep literature is recommended. Also, still pending to explain gap's reported in literature.

  1. I suggest to authors to bring link about algorithm proposed in order to replicate or reproduce it. In this manner is possible to know the applicability and pertinence about the algorithm proposed

  1. I recommend to authors to do a comparison with other method are suggested to validate the proposal. It is missing a section about comparisons.

  1. Also, an acronym list is recommended to explain in good manner

Reviewer 2 Report

The manuscript needs some considerable improvements,

What means software code e CVE-2021-3711?

the National Vulnerability Database? Which nation?

The acronyms should not be used in the title and the abstract without full wording when introduced.

Why  Petri nets were selected? The reasoning is weak, saying they are frequently used.

What is explicitly novel and original? How the results from this paper can be implemented in similar and/or other cases?

Multiple references are of no use for a reader and can substitute even a kind of plagiarism, as sometimes authors are using them without proper studies of all references used. In this case, each reference should be justified by it is used and at least a short assessment provided.

Please eliminate all multiple references. Please check the manuscript thoroughly and eliminate ALL the lumps in the manuscript. This should be done by characterising each reference individually. This can be done by mentioning 1 or 2 phrases per reference to show how it is different from the others and why it deserves mentioning.

43.6968ms - it should be a blank gap between numbers and symbols of units

43.6968 ms

In your conclusions, please discuss the implications of your research. Discussions and conclusions must go deeper, it would be more interesting if the authors focus more on the significance of their findings regarding the importance of the interrelationship between the obtained results and sustainable development in the sector context, and the barriers to doing it, what would be the consequences, in the real world, in changing the observed situation, what would be the ways, in the real world, to change/improve the observed situation.

Round 2

Reviewer 2 Report

Some small issues are still to be rectified e.g.

Shen J, etc[26]  should be Shen J, etc [26]

BTW what mean etc?

Should not be et al. ?

One more careful reading is needed.